# BCG-Vaccinated Children with Contact to Tuberculosis Patients Show Delayed Conversion of *Mycobacterium tuberculosis*-Specific IFN-γ Release

**DOI:** 10.3390/vaccines11040855

**Published:** 2023-04-17

**Authors:** Dorcas O. Owusu, Ernest Adankwah, Wilfred Aniagyei, Isaac Acheampong, Difery Minadzi, Augustine Yeboah, Joseph F. Arthur, Millicent Lamptey, Monika M. Vivekanandan, Mohammed K. Abass, Francis Kumbel, Francis Osei-Yeboah, Amidu Gawusu, Linda Batsa Debrah, Alexander Debrah, Ertan Mayatepek, Julia Seyfarth, Richard O. Phillips, Marc Jacobsen

**Affiliations:** 1Kumasi Centre for Collaborative Research in Tropical Medicine (KCCR), Kumasi 00233, Ghana; 2Department of Medical Diagnostics, College of Health Sciences, Kwame Nkrumah University of Science and Technology (KNUST), Kumasi 00233, Ghana; 3Agogo Presbyterian Hospital, Agogo, Ghana; 4St. Mathias Catholic Hospital, Yeji, Ghana; 5Atebubu District Hospital, Atebubu, Ghana; 6Sene West Health Directorate, Kwame Danso, Ghana; 7Department of General Pediatrics, Neonatology and Pediatric Cardiology, Medical Faculty, University Hospital Duesseldorf, Heinrich-Heine University, 40225 Duesseldorf, Germany; 8School of Medicine and Dentistry, College of Health Sciences, Kwame Nkrumah University of Science and Technology (KNUST), Kumasi 00233, Ghana

**Keywords:** BCG vaccination, IFN-γ, children, tuberculosis contact, *Mycobacterium tuberculosis* infection

## Abstract

*Mycobacterium (M.) bovis* BCG vaccination is recommended for healthy babies after birth in several countries with a high prevalence of tuberculosis, including Ghana. Previous studies showed that BCG vaccination prevents individuals from developing severe clinical manifestations of tuberculosis, but BCG vaccination effects on the induction of IFN-γ after *M. tuberculosis* infection have hardly been investigated. Here, we performed IFN-γ-based T-cell assays (i.e., IFN-γ Release Assay, IGRA; T-cell activation and maturation marker assay, TAM-TB) in children who had contact with index tuberculosis patients (contacts). These contacts were classified as either being BCG vaccinated at birth (*n* = 77) or non-BCG-vaccinated (*n* = 17) and were followed up at three timepoints for a period of one year to determine immune conversion after *M. tuberculosis* exposure and potential infection. At baseline and month 3, BCG-vaccinated contacts had significantly lower IFN-γ levels after stimulation with *M. tuberculosis*-specific proteins as compared to non-BCG-vaccinated contacts. This resulted in decreased proportions of positive IGRA results (BCG-vaccinated: 60% at baseline, 57% at month 3; non-BCG-vaccinated: 77% and 88%, respectively) at month 3. However, until month 12, immune conversion in BCG-vaccinated contacts led to balanced proportions in IGRA responders and IFN-γ expression between the study groups. TAM-TB assay analyses confirmed higher proportions of IFN-γ-positive T-cells in non-BCG-vaccinated contacts. Low proportions of CD38-positive *M. tuberculosis*-specific T-cells were only detected in non-BCG-vaccinated contacts at baseline. These results suggest that BCG vaccination causes delayed immune conversion as well as differences in the phenotype of *M. tuberculosis*-specific T-cells in BCG-vaccinated contacts of tuberculosis patients. These differences are immune biomarker candidates for protection against the development of severe clinical tuberculosis manifestations.

## 1. Introduction

With more than 9 million cases annually, tuberculosis is among the leading causes of death by an infectious disease in humans. However, this number outlines the overall incidence of infection only partially as the vast majority of individuals infected with *Mycobacterium (M.) tuberculosis* do not progress to acute disease (more than 90% of adults). However, these individuals remain latently *M. tuberculosis*-infected (LTBI) potentially for their entire lives and carry the inherent risk of developing tuberculosis disease. Children have a higher risk of disease progression after infection (up to 40% in toddlers and babies) and, in addition, are more likely to become infected when in contact with contagious tuberculosis patients [1]. In addition, especially young children have a high risk of developing severe tuberculosis manifestations [2]. Against this background, several countries with a high prevalence of tuberculosis, including Ghana, recommend vaccination using the live *M. bovis BCG* vaccine (BCG) at birth [3]. Beneficial effects of BCG vaccination against the development of severe disease manifestations have been found in previous studies, but efficacy varies across regions and populations [4]. The exact mechanisms underlying partially protective BCG-mediated effects remain elusive. Central to this knowledge gap is that reliable biomarkers of protective immunity against *M. tuberculosis* are missing. Although key factors of immune protection against *M. tuberculosis* (e.g., IFN-γ, CD4^+^ T helper cells) have been identified, these do not predict disease progression for LTBI [5]. Latent *M. tuberculosis* infection can be detected by immune-based tests (including the tuberculosis skin test (TST) and IFN-γ release assays (IGRAs)). Since environmental mycobacteria and BCG show immune cross-reactivity with *M. tuberculosis* antigens used for TST, IGRAs have been developed that apply single immunogenic proteins of *M. tuberculosis* (a.o., Early Secretes Antigenic Target (ESAT)-6, Culture Filtrate Protein (CFP)-10) to detect *M. tuberculosis* infection with high specificity and sensitivity. Importantly, these antigens ESAT-6 and CFP-10 are not present in the BCG genome and, hence, discrimination of BCG-vaccine-induced T-cell responses and T-cells specific for *M. tuberculosis* infection can be achieved. Approximately six weeks after *M. tuberculosis* infection, immune tests become positive based on a sensitized T-cell response against *M. tuberculosis* antigens. However, recent studies found consistently negative IGRA tests in individuals with a high probability of being infected with *M. tuberculosis* [6,7]. This phenomenon has been termed ‘resistance’ to infection with *M. tuberculosis*, and individuals exhibiting the phenomenon have been termed ‘resisters’ [6,7]. The mechanisms underlying this phenomenon are not known and, moreover, it remains unclear if this phenomenon indicates resistance or, e.g., a prolonged time frame preceding immune conversion.

The present study used a previously described modulated IGRA [8] to determine and quantify the sensitized T-cell response in children with confirmed contact to tuberculosis index patients (contacts). These contacts were subclassified as either BCG-vaccinated or non-BCG-vaccinated and were repeatedly seen at four timepoints during a one-year follow-up. In addition, the flow cytometry-based ‘T-cell activation and maturation marker’ (TAM-TB) assay was performed. This assay combines cell-specific analysis of IFN-γ expression with phenotype characterization. CD38 has been included as a marker of recent T-cell activation in this assay. Previous studies detected aberrant high CD38 expression on antigen-specific T-cells from acute tuberculosis patients [9]. The effects of BCG vaccination on IGRA conversion and antigen-specific T-cell phenotype were determined on this basis. 

## 2. Material and Methods

### 2.1. Study Cohorts and Clinical Characterization

We recruited asymptomatic contacts of *M. tuberculosis*-infected patients (contacts, *n* = 94) between April 2019 and February 2021 in Ghana. Age and sex characteristics of study groups are summarized in Table 1. The contacts recruited were associated with index tuberculosis patients attending one of four contributing hospitals (i.e., the Agogo Presbyterian Hospital, the St Mathias Catholic Hospital, the Atebubu District Hospital, and the Sene West District Hospital). The contacts were enrolled if they showed no clinical symptoms and no history of tuberculosis but were close relatives living in the same household with indexed tuberculosis patients. All study participants were screened for the presence of a BCG vaccination scar. Follow-up at month (M)3, M6, and M12 was conducted for all included contacts. None of the contacts progressed to tuberculosis during the study period. Study participants completed questionnaires to provide their demographic information, disease history, and period of exposure to the tuberculosis index patient. In addition, all participants were examined for other possible co-infections (i.e., *Schistosoma haematobium, Plasmodium falciparum, Giardia lamblia, Ascaris lumbricoides, Hookworm, Mansonella perstans*; Table 1). In this context, aliquots of blood, urine, and stool samples were analyzed microscopically for common infections.

The study obtained ethical approval from the Committee on Human Research, Publication and Ethics (CHRPE/AP/023/18) at the School of Medical Sciences at the Kwame Nkrumah University of Science and Technology (KNUST) in Kumasi, Ghana. All study participants and their legal guardians gave written informed consent prior to recruitment. Blood, urine, and stool samples were obtained from all study participants for laboratory analyses.

### 2.2. Whole Blood Stimulation Assays and Quantification of IFN-γ

Whole blood (100 µL/well) was mixed with cell culture medium (100 µL; RPMI supplemented with penicillin/streptomycin (100 U/mL) and L-glutamine (2 mM)) containing the different stimuli duplicates in individual wells. These stimuli comprised a recombinant ESAT6-CFP10 fusion protein (2 µg/mL) (kind gift of Prof. T. Ottenhoff, Leiden-University Medical Center), purified protein derivative of *Mtb* (PPD_Mtb_; 10 µg/mL; Statens Serum Institute, Copenhagen, Denmark), phytohemagglutinin (PHA; 10 µg/mL; Sigma-Aldrich, St. Louis, MO, USA), and a no stimulus control. Cell culture was carried out in 96-well U-bottom plates for 20 h at 37 °C and 5% CO_2_. The supernatants were harvested thereafter and stored at −80 °C until further analysis.

IFN-γ concentrations were measured using the Human IFN-γ Duo Set ELISA kit (R&D Systems, Minneapolis, MN, USA), following the manufacturer’s instructions. All samples were run in duplicates and analyzed using an Infinite M200 ELISA reader (Tecan, Zürich, Switzerland). Concentrations were determined from the standard curves using 4-parametric logistic regression. The IFN-γ concentrations of the non-stimulated samples were subtracted from the corresponding *Mtb* antigen-specific and PHA-induced IFN-γ values for final analyses. Values below the standard curve were set to 1 pg/mL for depiction and calculations. For the classification of test results as positive, negative, or indeterminate, we applied adjusted manufacturers’ criteria. ESAT6-CFP10 fusion protein was used for quantification and classification of the samples (as described [8]). In brief, a positive response was defined as an *M. tuberculosis* antigen-induced IFN-γ concentrations at least 25% and 17.5 pg/mL above the non-stimulated samples. IFN-γ concentrations above 400 pg/mL in the non-stimulated samples and/or less than 25 pg/mL in the PHA sample above the non-stimulated sample were classified as indeterminate.

### 2.3. T-Cell Activation Marker (TAM)-TB Assay

The TAM-TB assay was performed as previously described [9]. In brief, whole blood (100 µL/well) was mixed with cell culture medium (80 µL/well; see above) supplemented with co-stimulatory antibodies against human CD28 and CD49d (both 1 mg/mL; BioLegend). As for the IGRA assays, ESAT6-CFP10, PPD_Mtb_, and PHA were used for stimulation. After 2 h of cell culture, the Golgi inhibitor Brefeldin A (2.5 μg/mL) was added and, thereafter, the samples were cultured for an additional 18 h at 37 °C and 5% CO_2_.

For antibody staining, the plates were centrifuged (300 g; 5 min) and the supernatants were discarded. The sediments from the whole blood were then resuspended with fluorescence-labeled antibodies against human CD38 (APC; clone HIT2), CD4 (PerCP/Cy5.5; clone RPAT4), and CD3 (FITC; clone HIT3a, BioLegend) and incubated on ice in the dark (30 min). Thereafter, the erythrocytes were lysed (red blood cell lysis buffer, 100 μL/well; Sigma-Aldrich) following manufacturers’ instructions and the cells washed in sterile PBS (200 µL/well; Gibco). Samples were then fixed (Fixation buffer; BioLegend) in the dark at room temperature for 15 min and permeabilized (perm/wash buffer; Biolegend) according to the manufacturers’ instructions. The samples were then stained with IFN-γ (PE; clone B27, BioLegend) diluted in perm/wash buffer (5 µL/well). The plates were incubated in the dark at 4 ^°^C for 30 min and subsequently washed 2 times in perm/wash buffer. Stained cells were resuspended in PBS (100 µL/well) for measurement. Cells were acquired using BD Accuri C6 flow cytometer (BD Biosciences) and data were analyzed by FlowJo software (Version 10.8, FlowJo LLC, Ashland, OR, USA). Proportions of non-stimulated samples were subtracted from each stimulated condition and values below the estimated detection limit of 0.001% were set to 0.001% for depiction. Graphs were generated using GraphPad Prism version 9. A representative example gating procedure is provided as Appendix A. An arbitrary threshold of 0.03% IFN-γ producing CD4^+^ T-cells was set for classifying a contact as a positive TAM-TB response. Sample material was not sufficient for all participants to perform this assay and the respective numbers of included samples are provided in the figure legends. For analyses of CD38 expression, only participants with a positive ESAT-6/CFP-10-induced IFN-γ response (see above) at BL were included.

### 2.4. Statistics

All statistical analyses were performed using GraphPad Prism v9 software (GraphPad Software, La Jolla, CA, USA). Distribution tests (i.e., Kolmogorov–Smirnov, Shapiro–Wilk) were performed and did not suggest Gaussian distributions. In accordance, we used the non-parametric Mann–Whitney U-test for independent datasets. Non-continuous variables were compared among the study groups using Fisher’s exact test. A *p*-value below 0.05 was considered statistically significant. 

## 3. Results

In the context of a case/control study, we identified index tuberculosis patients (*n* = 32) and their potential contacts from children and adolescents living in the same household. A total number of 274 children and adolescents agreed to participate in this study, which included blood sampling at baseline (BL) and three follow-up timepoints (i.e., 3 months, M3; 6 months, M6; 12 months, M12 after BL). Only participants who completed follow-up at all timepoints were included (*n* = 94). BCG vaccination is recommended at birth for all newborns in Ghana. However, a considerable subgroup of participants had no BCG scar, the typical feature caused by BCG vaccination. According to BCG scar detection, these contacts were subclassified as BCG-vaccinated (*n* = 77) or non-BCG-vaccinated (*n* = 17). Initially, we compared these subgroups for age and sex distribution as well as for co-infections and other health issues (Table 1). No significant differences were found between BCG-vaccinated and non-BCG-vaccinated children for any parameter (Table 1). Moreover, the period of exposure to the respective index tuberculosis patient was determined, and although the non-BCG-vaccinated study group comprised higher proportions of individuals with 3 or 6 months of exposure, no significant differences were detected (*p* = 0.395) (Table 1).

Next, we performed whole blood IGRAs (comparable to QuantiFeron) that detect sensitized T-cell response against previous *M. tuberculosis* infection. IFN-γ concentrations in supernatants after in vitro culture with *M. tuberculosis* antigens were compared between the study groups. At BL, lower values for the study group of BCG-vaccinated contacts were detected for ESAT-6/CFP-10 and PPD_Mtb_ stimulations (*p* = 0.027 and 0.013, respectively; Table 2). Similarly, at M3, lower IFN-γ concentrations were found for both ESAT-6/CFP-10 and PPD_Mtb_ in BCG-vaccinated contacts (*p* = 0.009 and *p* = 0.007, respectively; Table 2). No differences were seen for the mitogen PHA at BL or M3 (Table 2). Although group median IFN-γ values were lower for both antigens in BCG-vaccinated contacts also at late timepoints, these differences reached significance only at the M12 timepoint (ESAT-6/CFP-10, *p* = 0.019; PPD_Mtb_: 0.028; Table 2). Based on adjusted manufacturers’ evaluation of IGRA test results (see methods), we next compared proportions of positive, negative, and indeterminate test results between the study groups (Figure 1). The BCG-vaccinated contacts had higher proportions of IGRA-negative results throughout the study timepoints and these differences were significant for the M3 timepoint (*p* = 0.049; Figure 1. Notably, during the time course, the proportions of positive IGRA results increased in BCG-vaccinated contacts and the study groups became more similar until M12 (BCG-vaccinated, 77%; non-BCG-vaccinated, 88%; Figure 1).

Furthermore, we compared both study groups for the phenotype of *M. tuberculosis*-specific T-cell responses using the TAM-TB assay. Lower proportions of both PPD_Mtb_ and ESAT-6/CFP-10-specific T-cells were found for the BCG-vaccinated study group at BL (ESAT-6/CFP-10, *p* = 0.007; PPD_Mtb_, *p* = 0.034; Figure 2) and this paralleled the IGRA results at this timepoint. Notably, no significant differences in the proportions of *M. tuberculosis*-specific IFN-γ-expressing CD4^+^ T-cells were detected at later timepoints (Figure 2). Using an arbitrary threshold of 0.03% IFN-γ-expressing ESAT6/CFP10-specific CD4^+^ T-cells, we compared the study groups for positive TAM-TB test results. We detected significantly higher proportions of positive TAM-TB results for non-BCG-vaccinated (40.0%) as compared to BCG-vaccinated (16.3%) children at BL (*p* = 0.032; Table 3). No significant differences were seen for the later timepoints.

Finally, we characterized the phenotype of IFN-γ-positive T-cell proportions for TAM-TB assay-positive individuals using the marker CD38, which was found to be a feature of *M. tuberculosis*-specific T-cells during acute tuberculosis [9]. In general, the population of CD38^+^ CD4^+^ T-cells producing IFN-γ was small and only the study group of non-BCG-vaccinated children had detectable proportions of CD38-positive *M. tuberculosis*-specific T-cells at BL (ESAT6/CFP10: 14.2%; PPD_Mtb_: 2%; Figure 3). Since only minor subsets from both contacts’ study groups had positive ESAT6/CFP10-induced IFN-γ responses (BCG-vaccinated, *n* = 9; non-BCG-vaccinated, *n* = 6), no statistical tests were performed.

## 4. Discussion

The present study characterizes the T-cell response to *M. tuberculosis*-specific antigens in healthy children as well as adolescents with contact to index patients with tuberculosis. All participants were classified as being BCG-vaccinated or non-BCG-vaccinated and these two study groups were compared throughout. Comparable exposure to *M. tuberculosis* was ensured for the subgroups to avoid a possible bias. Follow-up for one year, including repetitive analyses of immune response, demonstrated higher *M. tuberculosis*-specific IFN-γ expression and higher proportions of IGRA/TAM-TB-positive individuals in the non-BCG-vaccinated study groups at BL and month 3 (only for IGRA). The results are in accordance with a previous study by Verrall et al., where they found that BCG-vaccinated household contacts were less likely to be positive for IGRA tests at enrolment and had a lower risk for IGRA conversion after 14 weeks of follow-up [10]. As seen by others, the BCG effect declined with age, and no differences was observed in contacts older than 38 years [10,11]. In contrast, another study that investigated immune responses of contacts from a school outbreak after long-term exposure to an index tuberculosis patient found no association between BCG vaccination and IFN-γ release by ESAT-6-specific T-cells as measured by ELISPOT [12]. This fits well with our observation that a protective BCG effect is transient and can only be detected at an early timepoint after exposure to a patient with tuberculosis. 

In the present study, IFN-γ expression was determined as the gold standard for the detection of T-cell responses against tuberculosis. However, our previous research indicated that the use of alternative cytokines, such as IL-6, improved the sensitivity for detection of *M. tuberculosis* infection [8,13,14]. Hence, we cannot exclude that T-cells from BCG-vaccinated children produced alternative cytokines (including also T_H_2 or T_H_17 cytokines) and are not detected by performed analyses. In addition, alternative *M. tuberculosis* antigens associated with latent infection have been shown to be T-cell targets in tuberculosis patients and contacts [15,16]. Therefore, the dominant T-cell antigens ESAT-6/CFP-10 used in the present study may only partially reflect the immune response in contacts and differences between BCG-vaccinated and non-BCG-vaccinated could be due to differential *M. tuberculosis* antigen recognition.

We provided initial evidence that the T-cell responses were also qualitatively different between the study groups. Whereas BCG-vaccinated contacts had no *M. tuberculosis*-specific T-cells that expressed CD38, these T-cells were detected in non-BCG-vaccinated children. However, the conclusions drawn from CD38 phenotyping are limited since only few individuals from both study groups responded to ESAT-6/CFP-10 stimulation (threshold 0.03% of CD4^+^ T-cells; BCG-vaccinated, *n* = 9; non-BCG-vaccinated, *n* = 6). This rendered the application of statistical tests inappropriate. In addition, only a minor subset of *M. tuberculosis*-specific T-cells expressed CD38, and although these were only detected in non-BCG-vaccinated contacts, we would not deduce a role of CD38 in recent T-cell activation in the absence of acute tuberculosis. CD38 expression was previously identified as a marker of recent activation seen in *M. tuberculosis*-specific T-cells from acute patients with tuberculosis [17,18]. Our results did not suggest a role of CD38 up-regulation during *M. tuberculosis*-specific T-cell activation in the absence of acute tuberculosis disease.

A limitation of this study is the reliance on a BCG scar as an indicator of BCG vaccination. The BCG vaccination scar is a good surrogate marker for vaccination since it develops in 95% of vaccinated children, but it depends on correct injection techniques [19]. However, we cannot exclude that few of the BCG-scar-negative children/adolescents were BCG-vaccinated but did not develop a scar. There is ongoing debate as to whether BCG efficacy is seen in the small proportion of children who do not develop a scar [20]. This would yield biased study results due to false negative scar findings less likely.

During the time course of follow-up, T-cell conversion was seen in BCG-vaccinated individuals and this suggested that the causative mechanisms are active only for a limited time. We can only speculate about the nature of the underlying mechanisms, but a possible explanation would be training of the innate immune response by BCG vaccination. Such mechanisms have been described previously and are intensively studied [21,22]. Epigenetic modifications on the hematopoietic stem cell level by BCG have been described and this led to modified innate immune cells such as macrophages [23]. In addition, training of innate immune cells by BCG vaccination may induce memory beyond the adaptive immune system [22]. Against this background, future studies should determine if the delayed T-cell conversion in BCG-vaccinated individuals are associated with described changes in the innate immune compartment.

As stated before, none of the participants progressed to tuberculosis disease and, hence, we cannot conclude if the differences seen would also affect disease severity, as has been shown for BCG vaccination [4]. Larger study groups would be necessary to address this important question. Such studies will also investigate if the time frame until immune conversion, the phenotype of *M. tuberculosis*-specific T-cells, or other BCG-induced changes in innate immune response are biomarkers of disease severity. 

## 5. Conclusions

Our findings suggested that immune conversion towards recognition *M. tuberculosis* antigens happens earlier in non-BCG-vaccinated children and adolescents, implicating, e.g., an increased threshold for T-cell activation or other mechanisms relevant during early *M. tuberculosis* infections, which delay T-cell conversion in BCG-vaccinated individuals.

## Figures and Tables

**Figure 1 vaccines-11-00855-f001:**
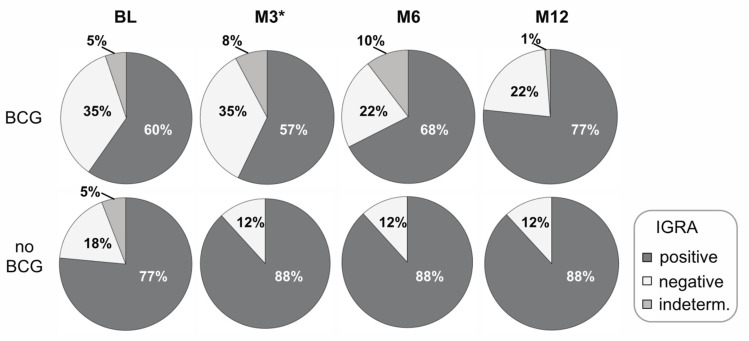
Time course comparison of BCG-vaccinated and non-BCG-vaccinated children with confirmed contact of tuberculosis patients for IGRA test results. Blood samples of contacts from tuberculosis index patients classified as either BCG-vaccinated (*n* = 77) or non-BCG-vaccinated (*n* = 17) compared for positive, negative, or intermediate IGRA results are shown as pie charts for four timepoints (i.e., BL, M3, M6, M12). The Fisher exact test was applied to analyze significance of differences between the study groups for individual timepoints. A *p*-value of <0.05 was considered significant and is indicated by *. indeterm., indeterminate.

**Figure 2 vaccines-11-00855-f002:**
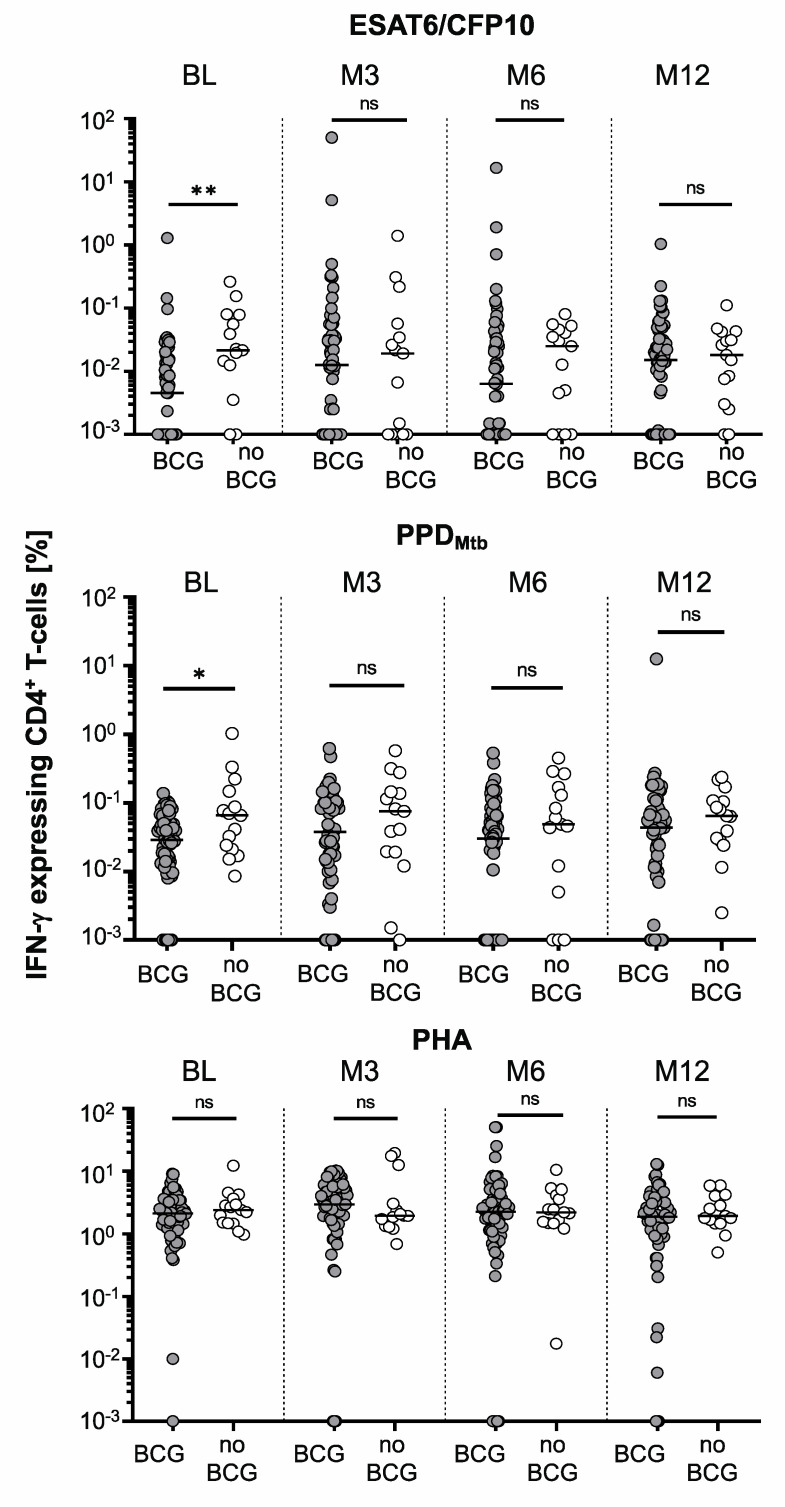
Comparison of *M. tuberculosis*-specific IFN-γ-positive CD4^+^ T-cell proportions between BCG-vaccinated and non-BCG-vaccinated children at BL and follow-up timepoints. Proportions of IFN-γ-positive CD4^+^ T-cells stimulated with M. tuberculosis antigens ESAT6/CFP10, PPD_Mtb_, or PHA are shown as symbol plots. Each circle indicates mean values of duplicate measurements for individual contacts with BCG vaccination (dark gray circles, *n* = 64) or without (open circles, *n* = 15). Solid lines indicate the median values for individual groups and timepoints. The Mann–Whitney U-test was applied and a *p* < 0.05 was considered statistically significant. Asterisks indicate significant differences (* for *p* < 0.05, ** for *p* < 0.01). ns, non-significant.

**Figure 3 vaccines-11-00855-f003:**
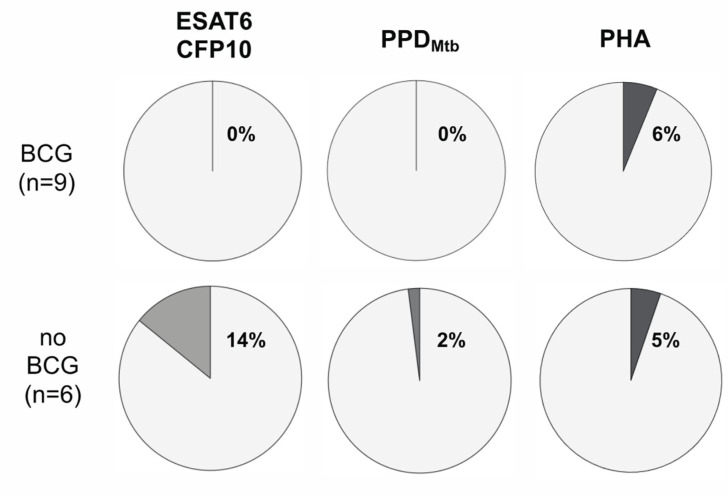
CD38 expression of *M. tuberculosis*-specific IFN-γ-positive CD4^+^ T-cell proportions between BCG-vaccinated and non-BCG-vaccinated children at BL. Proportions of CD38^+^ cells within the population of CD4^+^ IFN-γ-expressing ESAT6/CFP10-specific T-cells at BL are shown as pie charts for study groups of BCG-vaccinated (*n* = 9) and non-BCG-vaccinated (*n* = 6) children. Median values are depicted (dark gray for CD38 ^+^; bright gray for CD38^−^).

**Table 1 vaccines-11-00855-t001:** Characteristics of Study Groups.

	BCG Vaccination Status		
Characteristics	Yes (*n* = 77)	No (*n* = 17)	*p*-Value
**Age (median, range)**	10 (5–16)	10 (5–16)	0.419
**Sex_male,_ *n* (%)**	35 (45.5)	8 (47.1)	0.999
**Co-infections, *n* (%)**			
Schistosoma haematobium	2 (2.6)	1 (5.9)	0.454
Plasmodium falciparum	30 (37.0)	9 (52.9)	0.415
Giardia lamblia	2 (2.6)	1 (5.9)	0.454
Ascaris lumbricoides	5 (6.5)	0 (0)	0.581
Hookworm	3 (3.9)	0 (0)	0.999
Mansonella perstans	2 (2.6)	0 (0)	0.999
**Disease History, *n* (%)**			
Diabetes	2 (2.6)	0 (0)	0.999
Gastrointestinal diseases	1 (1.3)	0 (0)	0.999
Allergy	1 (1.3)	0 (0)	0.999
**Period of exposure to index tuberculosis patients, *n* (%)**			
1 month	4 (5.2)	0 (0)	0.395
2 months	31 (40.3)	5 (29.4)	
3 months	39 (50.6)	10 (58.8)	
6 months	3 (3.9)	2 (11.8)	

Abbreviations: Number, *n*.

**Table 2 vaccines-11-00855-t002:** IFN-γ concentrations after in vitro restimulation of blood samples from BCG-vaccinated and non-BCG-vaccinated children with *M. tuberculosis* antigens.

	BCG Vaccination Status		
Timepoint/ Stimulus	Yes (*n* = 77) median (range)	No. (*n* = 17) median (range)	*p*-Value
BL			
PHA	310 (0–4394)	191 (0–4025)	0.763
PPD	109 (0–4458)	359 (0–2310)	0.013
ESAT-6/CFP-10	41 (0–1870)	120 (0–1399)	0.027
M3			
PHA	276 (0–2499)	422 (43–3494)	0.304
PPD	72 (0–2572)	299 (26–2451)	0.007
ESAT-6/CFP-10	36 (0–1421)	142 (0–2361)	0.009
M6			
PHA	484 (0–5681)	358 (0–3698)	0.753
PPD	99 (0–5425)	217 (16–2629)	0.110
ESAT-6/CFP-10	86 (0–3963)	141 (0–5523)	0.190
M12			
PHA	476 (0–4358)	845 (0–2350)	0.598
PPD	149 (0–4722)	400 (0–1769)	0.019
ESAT-6/CFP-10	56 (0–5449)	148 (0–1243)	0.028

Abbreviations: Number, *n*; months, M.

**Table 3 vaccines-11-00855-t003:** Comparison of BCG-vaccinated and non-BCG-vaccinated children for TAM-TB-based diagnosis of *M. tuberculosis* infection.

TAM-TB	BCG TAM-TB-Positive, *n* (%)	Non-BCG TAM-TB-Positive, *n* (%)	*p*-Value
BL	9 (16.3)	6 (40.0)	0.032
M3	22 (34.4)	5 (33.3)	1
M6	16 (25.0)	7 (46.7)	0.119
M12	18 (28.1)	6 (40.0)	0.368

Abbreviations: Number, *n*; months, M.

## Data Availability

Any dataset is available from the authors upon reasonable request.

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
