# Peer review of "BCG-Vaccinated Children with Contact to Tuberculosis Patients Show Delayed Conversion of Mycobacterium tuberculosis-Specific IFN-γ Release"

_vaccines, 2023, doi:10.3390/vaccines11040855_

Round 1
Reviewer 1 Report
Major comments
This article describes the T-cell response to M. tuberculosis-specific antigens in healthy children and adolescents who were in contact with index patients with tuberculosis. The authors classified contacts as BCG vaccinated or not vaccinated.
The authors searched for differences in immune responses for the following items between the two groups. Quantitative INF-γ production and qualitative (number of positive people) evaluation of ESAT-6/CFP-10 and PPD were performed. CD38-positive T-cells (activation marker that is predominant in active tuberculosis) were detected.
This study helps to clarify the anti-tuberculous response in BCG recipients. The results of this study are interesting. We believe that the manuscript should be published.
However, the authors need to address the following issues.
1. The relationship between CD38 and the immunoreaction of BCG vaccine recipients is unclear. This aspect should be clarified.
Because CD38 is not released, is it not possible to protect it, or does it work defensively when it is released?
CD38 is also considered an indicator of tuberculosis activity.
The low number of positive cases among vaccinated people may be because the activity of tuberculosis was suppressed. Is this interpretation correct?
Pediatric tuberculosis progresses faster than that in adults, and symptoms develop in only 2 to 3 months after being infected by contact with the patient. In this period, CD38 levels in BCG-vaccinated contacts were lower than those in non-vaccinated contacts. Therefore, does this finding indicate that the infection has been suppressed and it shows an effect of BCG?
INF-γ is a cytokine that is important in killing tuberculosis bacteria, but excessive secretion is aggravating in infected individuals. Low secretion of INF-γ in BCG-vaccinated individuals may control inflammation due to infection.
Based on this information, I would like to ask you to consider whether BCG-vaccinated individuals also secrete cytokines that are included in TH2 cytokines among the alternative cytokines that the author refers to.
Hopefully, these things will be added to the discussion.
Minor comments
1. Regarding contact within the family, did the person who had the contact become infected? If PPD was negative, was infection established?
Should these people be excluded if the  immune reaction of infected people is investigated?
2. The subjects differ in the elapsed time from contact. Is it possible to evaluate blood samples with different exposure periods if the blood collection dates are the same?
Author Response
Major comments
This article describes the T-cell response to M. tuberculosis-specific antigens in healthy children and adolescents who were in contact with index patients with tuberculosis. The authors classified contacts as BCG vaccinated or not vaccinated.
The authors searched for differences in immune responses for the following items between the two groups. Quantitative INF-γ production and qualitative (number of positive people) evaluation of ESAT-6/CFP-10 and PPD were performed. CD38-positive T-cells (activation marker that is predominant in active tuberculosis) were detected.
This study helps to clarify the anti-tuberculous response in BCG recipients. The results of this study are interesting. We believe that the manuscript should be published.
However, the authors need to address the following issues:
The relationship between CD38 and the immunoreaction of BCG vaccine recipients is unclear.
This aspect should be clarified.
Response: CD38 has been identified as a marker of recent T-cell activation and its aberrant high expression on antigen-specific T cells was shown to be a feature of acute tuberculosis. We included CD38 for flow cytometry phenotyping in the present study to determine if CD38 expression is also detectable on recently activated T cells in the absence of disease progression.
The conclusions drawn from this analysis are limited since only few individuals from both study groups responded to ESAT-6/CFP-10 stimulation (threshold 0.03% of CD4+ T cells; BCG vaccinated, n=9; non-BCG vaccinated, n=6). This rendered the application of statistical tests inappropriate. In addition, only on a minor subset of M. tuberculosis specific T cells expressed CD38 and although these were only detected in non-BCG vaccinated contacts, we would not deduce a role of CD38 in recent T-cell activation in the absence of acute tuberculosis. We added this information to the introduction (p.4, l.9 to 12), the results (p.11, l.2 to 4) and the discussion section (p.12, l.23 to 26; p.13, l.1 to 4) of the revised manuscript.
Because CD38 is not released, is it not possible to protect it, or does it work defensively when it is released?
Response: As stated above only membrane-associated CD38 was determined as a marker of antigen specific T cells.
CD38 is also considered an indicator of tuberculosis activity.
Response: This was the background for including CD38 as a phenotypic marker. However, low proportions CD38 positive T cells (only detectable for ESAT6-CFP-10 specific T cells of non-BCG vaccinated) may indicate that CD38 is predominantly expressed in acute disease. We added this information to the discussion section (p.12, l.23 to 26; p.13, l.1 to 7) of the revised manuscript.
The low number of positive cases among vaccinated people may be because the activity of tuberculosis was suppressed. Is this interpretation correct?
Response: This is one possible explanation. Alternatively. one may speculate that immunopathology of acute tuberculosis is causative for CD38 expression. We added this point to the discussion section of the revised manuscript (p.12, l.25 to 26).
Pediatric tuberculosis progresses faster than that in adults, and symptoms develop in only 2 to 3 months after being infected by contact with the patient. In this period, CD38 levels in BCG-vaccinated contacts were lower than those in non-vaccinated contacts. Therefore, does this finding indicate that the infection has been suppressed and it shows an effect of BCG?
Response: In the absence of statistical significance of the findings concerning CD38, we would not speculate on the clinical relevance.
INF-γ is a cytokine that is important in killing tuberculosis bacteria, but excessive secretion is aggravating in infected individuals. Low secretion of INF-γ in BCG-vaccinated individuals may control inflammation due to infection.
Response: This is an important point. We totally agree that a balanced immune response is decisive for immune protection against M. tuberculosis. This assumption is strengthened by the fact that high IFN-g expression by T cells and in the serum are seen in tuberculosis patients.
Based on this information, I would like to ask you to consider whether BCG-vaccinated individuals also secrete cytokines that are included in TH2 cytokines among the alternative cytokines that the author refers to.
Response: We agree that future studies should include TH2 cytokine analyses and we added this point to the discussion section of the revised manuscript (p.13, l.12 to 13).
Hopefully, these things will be added to the discussion.
Minor comments
Regarding contact within the family, did the person who had the contact become infected? If PPD was negative, was infection established? Should these people be excluded if the immune reaction of infected people is investigated?
Response: All study participants are contacts (i.e., family members and/or household members) of confirmed tuberculosis patients. Some participants showed M. tuberculosis specific T-cell responses at baseline and others converted during the time course. We tried to improve the clarity of the revised manuscript in this regard.
The subjects differ in the elapsed time from contact. Is it possible to evaluate blood samples with different exposure periods if the blood collection dates are the same?
Response: Although we cannot exclude effects of exposure time, our analyses did not suggest significant differences between the study groups in this regard (p.9, l.14 to 17).
Reviewer 2 Report
I think the study is of sufficient novelty to merit publication. However, the draft requires significant editing to improve the quality of presentation, type-setting and English. Figure 1 depicting the results on which the whole publication hinges is too small and hard to read. In its present format it lacks impact and I suggest splitting it out into three different figures of good size and quality. There are gaps in the literature awareness in the introduction and elsewhere in the body text. I suggest these are updated. I attach a scanned hard copy with my suggested amendments.

Author Response
I think the study is of sufficient novelty to merit publication. However, the draft requires significant editing to improve the quality of presentation, type-setting and English.
Response: We revised the manuscript accordingly to improve the quality.
Figure 1 depicting the results on which the whole publication hinges is too small and hard to read. In its present format it lacks impact and I suggest splitting it out into three different figures of good size and quality.
Response: We followed this reviewer suggestion and split Figure 1a, 1b, and 1c into Figure 1, 2, and 3 in the revised manuscript.
There are gaps in the literature awareness in the introduction and elsewhere in the body text. I suggest these are updated.
Response: We followed two of the doi links (one was incomplete) included by the reviewer and noticed that the review articles were about vaccination effects on innate immunity training. Although this topic is of great relevance, it is not addressed in the present study and it does not add to the point if BCG vaccination can protect against tuberculosis disease progression. We assume that we misunderstood the reviewer and, maybe, there are original articles, which show such effects. If this is the case, we will be happy to include.
I attach a scanned hard copy with my suggested amendments.
Response: We included all changes we could decipher and in case these are reasonable.